# Mapping Ecosystem Services Bundles for Spatial Planning with the AHP Technique: A Case Study in Tuscany (Italy)

**Massimo Rovai** [1,*] , **Tommaso Trinchetti** [2] , **Francesco Monacci** [2] **and Maria Andreoli** [3]

1 Department of Civil and Industrial Engineering (DICI), University of Pisa, 56122 Pisa, Italy
2 Department of Agricultural, Food and Agro-Environmental Sciences (DAFE), University of Pisa, 56124 Pisa, Italy; tommaso.trinchetti@phd.unipi.it (T.T.); francesco.monacci@gmail.com (F.M.)
3 Independent Researcher, 56124 Pisa, Italy; maria.andreoli.jussila@gmail.com
* Correspondence: massimo.rovai@unipi.it

**Abstract:** Agricultural and forest ecosystems provide multiple ecosystem services (ESs) fundamental to the well-being and quality of life of citizens. However, in the European context, these ecosystems are often threatened by processes of urban development, around cities, or abandonment, in mountainous or remote areas. Faced with the need for solutions oriented towards greater sustainability and resilience of socio-ecological systems, planning should contribute to rebuilding more integrated and mutually beneficial relationships between urban and rural areas, ensuring the effective production of multiple ESs. The regulation and management of ESs are complex and require scientifically sound and widely understandable policies and governance models, based on detailed assessment methods. This paper proposes a method for mapping and bundling the supply of five ESs produced in agricultural and forest areas, based on the processing of open source territorial data through the analytic hierarchy process (AHP), and tailored for the Tuscany region (Italy). The method integrates the land use and land cover map with other data to obtain a comprehensive ESs assessment, and then uses cluster analysis to identify bundles of ESs. Based on a first trial, the method seems to show high potentialities as a Decision Support System to promote innovative governance models for ES management.

**Keywords:** ecosystem services mapping; AHP; rural areas; Tuscany



## 1. Introduction

Land use changes are factors with the highest impacts on biodiversity and ecosystems worldwide [1]. In Europe, the analysis of land use changes highlights two main trends: urban development and the abandonment of cultivated land [2].

Since the middle of last century, the rapid economic, social, and institutional changes that occurred in most developed regions of Europe have fostered huge development in urban areas, mainly characterized by dispersion of low-density settlements around city centers, albeit with different specificities linked to local socio-economic contexts [3]. This phenomenon has caused several negative consequences, including the loss and fragmentation of peri-urban agricultural areas, often those with the most fertile soils, a reduction in food production and a dependence on imports [4].

At the same time, many regions of Europe have experienced the progressive abandonment of cultivated land in the more remote and mountainous areas which are less productive and profitable due to the loss of competitiveness on food markets [5]. The consequences of this phenomenon are contrasting and debated [6], and strongly depend on local socio-environmental conditions and on policies promoted for its management [7]. Some authors highlight the advantages of renaturalization, such as the creation of habitats for large mammals and an increased carbon sequestration [8]. Others underline the disadvantages caused by a lack of human intervention, e.g., large losses in food production, landscapes and cultural resources, and an increase in hydrogeological and fire risks [9].

Moreover, the widespread abandonment of cultivated land in Europe's remote areas adds to and increases the pressures on the remaining and more productive agricultural areas, both in European central and well-connected plains or in other regions of the world [10]. In fact, in the more fertile areas in lowlands and near urban centers that are still cultivated, agricultural production is often pursued through intensive practices, which impacts biodiversity and ecosystems. In addition, the impacts of intensive agriculture are also caused abroad, i.e., in developing countries, as a result of an increased demand for imported products [11].

In summary, urban development, the abandonment of cultivated land, and agricultural intensification are the major threats to the health and stability of ecosystems and should be considered as incorrect and unbalanced land management types [12]. Nowadays, difficulties in managing increasingly large and dispersed settlements, and their environmental impacts at local and global scales, necessitate the need to rebuild relationships between urban and rural areas to improve well-being and quality of life through the efficient use of resources and ecosystem services (ESs) produced by rural areas.

Increasing pressures on land and water compromise the productivity of world agriculture, while future food production will depend on the preservation and careful management of these resources [13]. The urgent need to reduce the environmental impacts of agriculture and, at the same time, to guarantee food security is at the base of a debate which often remains stuck in the contrast between food production and the maintenance of natural habitats. Vice versa, it is necessary to recognize the potential of agriculture to produce a variety of ESs that can support human well-being in many ways [14]. Research has identified positive relationships between multifunctional agriculture and the supply of ESs [15] both in urban [16] and rural areas [17].

The management of multiple ESs produced by agriculture, forestry and other natural areas is gaining increasing importance in political debates. An appropriate balance at a local level between the supply and demand of resources and ESs is essential for sustainability [18]. However, the fact that ESs are public goods produced by private actors determines difficulties in their promotion also due to the lack of adequate incentives [19]. The supply of these public goods is often not aligned with demand and political and governance measures are necessary for an adequate supply [20]. The design of strategies and measures for organizing the interaction between different levels, sectors, actors, and interests, to improve and make mutually beneficial the relations between urban, peri-urban, and rural areas, and to promote equilibrium and integration at the territorial level, is considered increasingly urgent [21].

Information and decision support systems (DSS) allow planners to use the available information on the structure and functions of ecosystems to assess the availability of ESs over space and time, to understand the most relevant ESs within a given context, and to favor interventions aimed to guarantee and/or improve supply at different scales (farms, districts and basins). In the last twenty-five years, research on ES assessment and mapping has significantly increased [22], and there have been many efforts to apply the results in policies and planning [23], including through appropriate DSS [24]. The availability of spatially explicit information is particularly important in the design and implementation of plans and policies for ES management, and many approaches and methods for mapping ESs have been developed [25]. The primary data can be used for provisioning services when statistics are available. When the primary data are not available, as often happens, especially for regulating and cultural services, ES mapping must be based on proxy data. In this case, the proposed approaches range from the simple use of land use and land cover (LULC) maps [26] to dynamic process-based models such as those available in the InVEST 3.13 [27], AIRES [28], or SolVES 4.1 software [29].

LULC maps combined with a "capacity matrix" are among the most widely used proxy methods, especially in planning. With this approach, the capacity of different areas of a territory to provide ESs is described in relative terms. In the matrix, scores are assigned to each land cover class based on expert judgment [30]. This method is very easy and quick

to apply and guarantees replicability and comparability in space and time. However, some critical issues emerge, related to the difficulty of describing the production of some ESs only based on land cover. In territorial contexts where other environmental, socio-economic, and geographical data are available, it may be interesting to investigate how to combine LULC maps with additional data by using multi-criteria analysis techniques, to obtain a better assessment of the capacity to provide ESs. Such techniques, in our opinion, can allow to develop methods for ES mapping that, from a methodological point of view, are not as resource-intensive as many models but, at the same time, can be more comprehensive and specific than only using LULC maps.

In this paper, we propose a method for mapping the ESs provided by agricultural and forest areas that uses the analytic hierarchy process (AHP) technique, and open source territorial data on properties and conditions of soils and ecosystems.

Furthermore, following the growing interest in the ability of ecosystems to simultaneously produce multiple ESs [31], research has attempted to investigate the relationships in time and space between the ESs and the trade-offs and synergies between them [32,33]. A better understanding of the relationships between different ESs could allow for the development of policy and governance measures oriented to manage a set of ESs, according to an integrated approach, and not according to each of them individually. Such an approach would make it possible to overcome the logic of sectoral policies that frequently come into conflict with each other. It could also help to avoid trade-offs and enhance synergies between the ESs, thus improving the overall resilience of ecosystems.

Recently, research has been able to identify and map the co-occurrence or association of ESs, which can give a first indication of their relationships, by using a methodology based on cluster analysis that allows to characterize a territory with ES bundles repeated in space and time [34]. Many different approaches for defining ES bundles have been developed, which vary for ESs considered, assessment methods, scale, and resolution. In any case, the possibility of analyzing the capacity of a territory to produce not just one but several ESs at the same time has proven to be important to inform policy and governance measures [35].

In this paper, we present the results of an ES mapping and bundling method tested on Lucchesia, i.e., a specific area located in the north-western part of the Tuscany region, Italy. Lucchesia is one of the twenty homogeneous areas called "Ambiti Paesaggistici" (Landscape Areas) identified by the main regional planning tool, the "Piano di Indirizzo Territoriale con valenza di Piano Paesaggistico Regionale" (PIT-PPR). The proposed method can represent a useful innovation for DSS.

The next section describes the study area, applied technique, used data, assessed ESs, and performed processing. The following sections contain results, their discussion, and conclusions.

## 2. Materials and Methods

### 2.1. Study Area

Tuscany is one of the 20 administrative regions of Italy and is in the central-western part of the country (Figure 1). It is a typical Mediterranean region characterized by a great variety of territorial and landscape contexts and has a good international reputation due to its landscapes [36] and its high-quality agricultural products. Therefore, the maintenance and enhancement of rural areas and the ESs they provide is a fundamental objective to be pursued. For this purpose, it is necessary to develop adequate IT tools for the assessment, planning and monitoring of ESs production in rural areas.

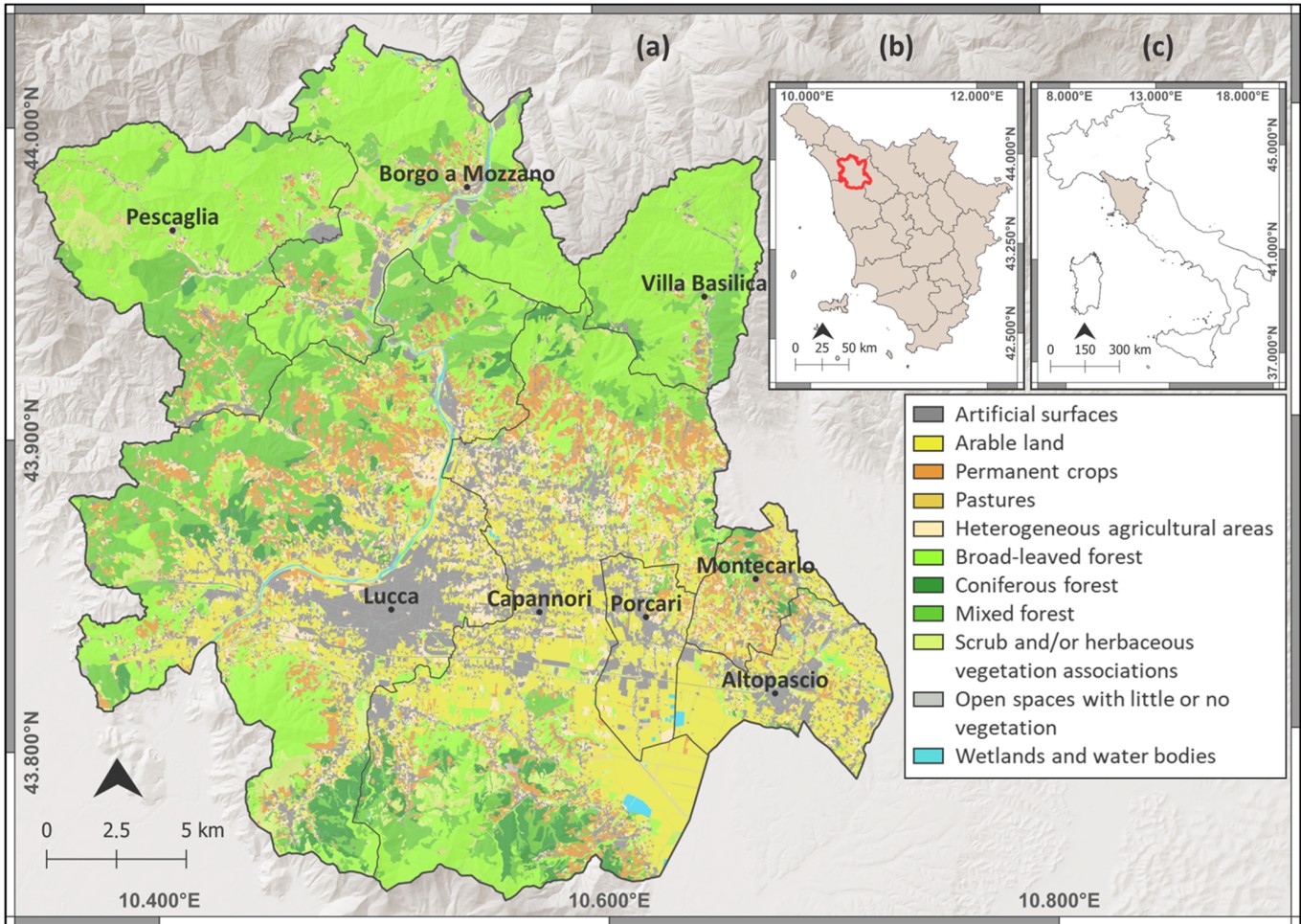

**Figure 1.** (**a**) Study area's municipal boundaries and land cover distribution in 2019. (**b**) Tuscany's Landscape Areas (black lines) and study area's location (red line). (**c**) Tuscany's location (national boundaries from ISTAT; regional and municipal boundaries from Tuscany Region; Landscape Areas boundaries from Tuscany PIT-PPR cartography).

The Tuscan Regional Law on the Government of the Territory n. 65/2014 aims to enhance territorial heritage, forwarding sustainable development, contrasting land consumption, and promoting multifunctionality, but integrated political and governance measures to promote specific protection and improvement actions still need to be adopted. Currently, between the different planning levels (i.e., regional and municipal), there are sometimes inconsistencies and conflicts in spatial development strategies [37]. For instance, at a municipal level, the demand for urban development and land consumption is still very high despite the regional government being strongly oriented towards soil and landscape protection.

The PIT-PPR states that territorial development objectives and strategies cannot conflict with the protection and enhancement of the landscape heritage. Furthermore, all regional sector policies, and all provincial and municipal planning tools, must be drafted in coherence with the PIT-PPR. It has classified the regional territory into twenty homogeneous Landscape Areas according to their historical and cultural identity, their rural, ecosystemic, and hydro-geomorphological characteristics, as well as to their socio-economic characteristics and to the characteristics of their settlement and infrastructural systems. The Landscape Areas have been considered the most appropriate territorial scale for ES mapping for our analysis. In fact, the supply of ESs, ensured by the complexity of ecosystems, is influenced by natural processes which, in turn, depend on the biophysical characteristics of a territory and not so much on administrative divisions [38].

Our method for mapping and bundling ESs allows us to create maps for all the twenty Landscape Areas of Tuscany and to make comparisons but, in this paper, we describe its application to the case study of the Lucchesia Landscape Area.

The territory of the Lucchesia is in the north-western part of Tuscany and has an extension of approx. 583 km$^2$, consisting of an extensive plain (66.7%), surrounded to the north, west and south by hills (17.9%) and mountains (15.4%) (Figure 1). It includes eight municipalities with a population of about 174,500 inhabitants, mainly concentrated in the plain, in the urban and peri-urban areas of Lucca (88,770) and Capannori (45,280). In 2019, Lucchesia's land cover distribution was as follows: artificial surfaces, 14.8%; agricultural areas, 31.2%; natural and semi-natural areas, 52.3%; water bodies and wetlands, 0.6%. Artificial surfaces accounted only for 2.1% of the mountains and 3.6% of the hills, but their share goes up to 20.7% in the plain. Agricultural areas were also concentrated in the plain, of which they represented 44.4%, while they were only 7.5% of the hills and 1.7% of the mountains. Natural and semi-natural areas dominated hills (88.7%) and mountains (96.2%) and represented 34.0% of the plain. The main critical factors present in the Landscape Area include:

- Abandonment of cultivated land and the lack of maintenance of forests in hills and mountains, which cause environmental and safety problems (e.g., increased hydrogeological and fire risks);
- Land consumption in the plain due to the growth of dispersed urban areas, which causes the loss and fragmentation of agricultural areas.

Planning is now looking for ways to limit agricultural land consumption and, at the same time, promote multifunctional agriculture for restoring value to rural areas. Therefore, the use of DSS to increase knowledge about the value of the ESs provided by the rural territory is fundamental for making decisions on spatial planning.

### 2.2. The Analytic Hierarchy Process (AHP) Technique

The method for ES mapping proposed in this paper uses the AHP, a multi-criteria analysis technique [39], and the software QGis 3.24. The integration of multi-criteria analysis techniques and GIS software is often used to create DSS tools in which the spatial component has a fundamental relevance [40], as in the case of ES assessment. In general, with multi-criteria analysis techniques, a set of relevant criteria is identified for comparing, evaluating, and ordering different alternatives. The chosen criteria are weighted according to the preferences of the decision-makers. Likewise, the alternatives are also weighted for each criterion. Once an ordering of the criteria and an ordering of the alternatives for each criterion have been obtained, they are re-aggregated into a single general ordering of preference of alternatives. Different multi-criteria analysis techniques exist [41]. A fundamental aspect of multi-criteria analysis is the possibility of evaluating alternatives both qualitatively and quantitatively, and with the unit of measure most suited to what the criteria describe, and this makes it particularly effective in addressing the complex and transdisciplinary problems of territorial planning [42].

Among the many multi-criteria analysis techniques, the AHP is widely used in spatial analyses. When integrating this technique with GIS, the alternatives are the single spatial units of a map, i.e., polygons of a vector file or pixels of a raster file [43]. In our case, to determine the capacity of each spatial unit (alternative) to provide an ES, the AHP technique requires the definition of a hierarchical tree which consists of three levels: objective, criteria and sub-criteria, and attributes. The objective, in our case, is the capacity to provide a specific ES. The criteria and sub-criteria are the characters or properties that are relevant to define the objective. The attributes are the values that alternatives can assume for each criterion or sub-criterion. Using a pairwise comparison matrix, weights are assigned to the elements of the hierarchical tree in the following order: criteria with respect to the objective; sub-criteria with respect to the criteria; attributes with respect to the sub-criteria. The hierarchical tree is then recomposed considering the weights assigned to attributes, sub-criteria, and criteria. In this way, for each alternative (spatial unit) it is possible to

determine a specific value for the objective (ES provided) on a relative dimensionless scale from 0 to 1. In conclusion, with this technique, which uses an equal scale of values for all the ESs, it is possible to create maps that allow to compare the capacity of different areas of the territory to provide the ESs.

Like many other multi-criteria analysis techniques, the AHP can be used by involving citizens (and/or stakeholders) in defining the weights of criteria, sub-criteria, and attributes. This encourages the growth of awareness about sustainability issues and to have indications on the priority of intervention. In this first analysis, authors have assigned weights using the paired comparison technique, a simplified method compared to the scale originally proposed by Saaty [39].

The next three sections explain the used data, the design of the hierarchical trees, and the processing performed to obtain the ES maps, respectively. The complete hierarchical trees for each ES Table S1 and weights assigned Table S2 are reported in the Supplementary Materials. For a more extensive explanation of the proposed method, see [44].

*2.3. Data*

In recent years, the Regional Government of Tuscany has created an open source territorial and environmental information system to make available geographic data that, among other things, can also be used for ES mapping. Data used in our ES assessment come from the Tuscany Region webgis, from the National webgis, and from the ISPRA (Higher Institute for Protection and Environmental Research, Rome, Italy) website.

2.3.1. Land Use and Land Cover Map

In 2007, the Tuscany region launched an LULC mapping program (scale 1:10,000) for the entire territory to update every three years. For our ES mapping, the LULC maps of the years 2016 and 2019 were used. As our analysis is limited to the ES provided by agricultural and forest areas, the elements of the two LULC maps with the codes of the agricultural and forest areas were selected and exported. In particular:

- Agricultural areas, corresponding to class 2 of the first level of the Corine Land Cover classification (CLC), and pastures, corresponding to class 321 of the third CLC classification level;
- Forest areas, corresponding to class 31 of the second CLC classification level and to class 324 of the third CLC classification level.

2.3.2. ARTEA Database

ARTEA is the regional agency that authorizes and controls the EU aid requested by farms under the European Agricultural Fund for Rural Development. To obtain it, farmers must submit the graphic cultivation plans (GCPs) of their farms in a digital format, indicating the spatialized cadastral references of the cultivated parcels, the crops, the ecological focus areas (EFAs) as well as the cultivation method (i.e., conventional, organic or in conversion).

Every year ARTEA makes the GCPs of all Tuscan farms available in an anonymous form. This dataset constitutes a very useful information base on Tuscan regional agriculture, from which we extracted and used the GCPs of the years from 2016 to 2020. The GCPs were overlaid on the LULC maps to identify the agricultural and forest areas managed by professional farms (the areas corresponding to the GCPs) and the agricultural and forest areas where hobby or residual agriculture is practiced. The GCPs of the years 2016/2017 were overlaid on the selected elements of the 2016 LULC map, while the GCPs of the years 2018/2020 on those of the 2019 LULC map. In this five-year period, a very high number of crops were cultivated in Tuscany making it necessary for reclassification by homogeneous groups. Three different classifications of crops were created. Firstly, crops were reclassified according to their main destination (Table 1). Secondly, they were reclassified according to the crop type, i.e., forests, permanent crops, pastures, "miglioratrici" crops, "rinnovo" crops,

or "depauperanti" crops. This classification, that considers the effects of crops on soil fertility, is a fundamental distinction for Italian agronomy (we use Italian terms for not having found a corresponding classification in English). Thirdly, crops were reclassified according to the length of their cycle, i.e., forests, permanent crops, poly annual, annual or seasonal crops. These three different classifications were used to evaluate the ability of crops to contribute to different ESs. Furthermore, the information present in the dataset made it possible to identify the cadastral parcels considered as EFAs (for example, terraces, hedges, rows of trees, groves, buffer strips).

**Table 1.** Crop main destination.

| Crops for Food | Crops for Animal Rearing | Crops for Fibers and Materials | Crops for Various Destinations |
|---|---|---|---|
| Horticulturals | Cereals for fodder | Forests | Arable crops |
| Cereals | Leguminous for fodder | Arboriculture | Greenhouses |
| Leguminous | Fodder | Nurseries | Heterogeneous areas |
| Orchards | Pastures | Oleaginous | |
| Vineyards | Grasslands | Ornamentals | |
| Olive groves | Retired land | Officinals | |

### 2.3.3. Pedological Database

In 2014, the LaMMA Consortium (Environmental Monitoring and Modeling Laboratory, Florence, Italy) created a Tuscany Land Capability map, integrating a photo interpretation process with the description and analysis of about 500 units of soils [45].

Land capability classification (LCC) includes two levels: classes and subclasses. The eight classes, numbered with Roman numerals, express soil potential for agricultural use for all possible and practicable crops, by considering the number and severity of limitations. Four subclasses indicate the type of limitation: soil properties, water excess, erosion risk, or climatic factors. Soils are classified according to the main physical and chemical properties (e.g., depth, texture, skeleton, organic matter, salinity); drainage and flood risk; slope and potential erosion; freezing or water shortage risk. To assess soil contribution to the provision of ESs, the LCC and all data on soil properties were used.

### 2.3.4. Nature Map

ISPRA leads the Nature map project to analyze the conditions of the environment at national and regional level, highlighting values and criticalities, a task required by the Framework Law on Protected Areas 394/1991. The project included a first phase of identification of habitats, related to land cover and lithological and geomorphological characteristics, according to CORINE Biotopes Classification, and a second phase of assessment, in terms of ecological value, ecological sensitivity, and anthropogenic pressure [46]. The Nature map of the Tuscan territory (scale 1:50,000) was used as an optimal reference for assessing the properties and conditions of ecosystems.

### 2.3.5. Cartography of the PIT-PPR

The PIT-PPR of Tuscany analyzed and interpreted the landscape related to four aspects (structural invariants): hydro-geomorphology; ecosystemic characteristics; settlement and infrastructural systems; rural landscapes characteristics. Maps for each of these structural invariants were produced together with the plan [47]. For ES mapping, the map of the regional Hydrographic Network and the map of the structural and functional elements of the regional Ecological Network were used.

### 2.3.6. Other Data

Other used data were: Orographic DTM; Regional Landslide Database; map of Landslide Hazard from River Basin Plan (PAI), updated in 2017 by ISPRA; map of areas subject to Hydrogeological Constraint (Law 3267/23); map of Annual Average Precipitation for period 1995–2014, provided by LaMMA Consortium; map of Inland Water Bodies; map of Hydraulic Hazard from Flood Risk Management Plan (PGRA), updated in 2021 by ISPRA; maps of Protected Areas.

### 2.4. Hierarchical Trees for Ecosystem Services Mapping

ESs are defined as benefits people obtain from ecosystems [48] and are realized when people use ecosystem functions, i.e., when ES supply meets people's demand [18]. It is important to distinguish between ES flows, which are dynamic, and ES stocks, considered as the capacity of the ecosystems to produce benefits, which is spatially localized. ES flows are difficult to capture on maps, while ES stocks are easier to map [49]. In detail, ES production is realized in the context of different components, which are interconnected, but can be mapped separately. Ecosystem properties and conditions are the basis for ES potential, which, together with human inputs, determines ES supply, i.e., the capacity of a socio-ecological system to produce ESs. ES flows can be a fraction of this supply, or be higher, depending on ES demand [50].

The sustainability of an ecosystem, and, therefore, of a territory, is based on the balance between supply and demand of ESs. Management difficulties emerge from the fact that demand is usually concentrated in some small spaces (e.g., urban areas) that are often far from places of production/supply of ESs [51].

The proposed method aims to map the supply of five ESs, selected with reference to the Common International Classification of Ecosystem Services (CICES) [52]. The CICES describes 72 ESs and assigns each of them a unique four-digit code. It allows them to be aggregated according to the thematic scope and spatial scale of the analysis. For our analysis, based on the characteristics of the Tuscan territory, it was decided to map four of the provisioning and six of the regulating ESs classified by the CICES, considered strategic for territorial and landscape planning. The selected provisioning ESs cover the main nutritional, non-nutritional materials, and energetic outputs from agricultural and forest ecosystems. They are aggregated in a single ES. The selected regulating ESs cover some of the ways in which the same ecosystems contribute to mediate or moderate the physical, chemical, and biological conditions of the environment. They are essential to ensure the stability and continuity of provisioning services, as well as human health, safety, and comfort. Two pairs of them are aggregated in two single ESs. In summary, the mapped ESs are the following:

- Provisioning of food (1.1.1.1), fibers and other materials (1.1.1.2), plants for energy (1.1.1.3), and reared animals (1.1.3.1), aggregated into a single service;
- Erosion control (2.2.1.1) and attenuation of mass movements (2.2.1.2), aggregated into a single service;
- Hydrological cycle and water flows regulation (2.2.1.3);
- Maintenance of habitats (2.2.2.3);
- Weathering processes (2.2.4.1) and decomposition processes (2.2.4.2) for soil quality, aggregated into a single service.

The aim of the proposed method is to map the capacity of each area of the territory to provide these five ESs on a relative scale. For each ES, the method uses a set of criteria for assessing soils and ecosystems properties and conditions, and a set of criteria for assessing human inputs, i.e., human contributions to land management in the five-year period 2016/2020 (Table 2).

**Table 2.** Hierarchical trees for ecosystem services mapping.

| Objective (ES) | Criteria and Weights | | | |
|---|---|---|---|---|
| Provisioning | Potential | 0.500 | Land capability | 1.000 |
| | Human inputs | 0.500 | Crop destination | 0.667 |
| | | | Type of activity | 0.333 |
| Soil quality | Potential | 0.667 | Soil physical properties | 0.417 |
| | | | Soil chemical properties | 0.417 |
| | | | Soil hydraulic properties | 0.167 |
| | Human inputs | 0.333 | Crop type | 0.500 |
| | | | Type of activity | 0.333 |
| | | | EFAs | 0.167 |
| Erosion and mass movements control | Potential | 0.667 | Soil vulnerability | 0.667 |
| | | | Soil physical properties | 0.333 |
| | Human inputs | 0.333 | Hydrogeological constraint | 0.100 |
| | | | Length of crop cycle | 0.400 |
| | | | Type of activity | 0.300 |
| | | | EFAs | 0.200 |
| Water flows regulation | Potential | 0.667 | Soil moisture | 0.300 |
| | | | Soil hydraulic properties | 0.300 |
| | | | Soil physical properties | 0.200 |
| | | | Distance from surface waters | 0.200 |
| | Human inputs | 0.333 | Length of crop cycle | 0.500 |
| | | | Type of activity | 0.333 |
| | | | EFAs | 0.167 |
| Maintenance of habitats | Potential | 0.667 | Ecological value | 0.500 |
| | | | Habitats vulnerability | 0.333 |
| | | | Ecological network | 0.167 |
| | Human inputs | 0.333 | Protected areas | 0.100 |
| | | | Crop type | 0.400 |
| | | | Type of activity | 0.300 |
| | | | EFAs | 0.200 |

### 2.4.1. Provisioning

The production of food and other materials used by people is a fundamental service, which is often mapped. Typically, for mapping this service, an LULC map combined with agricultural production estimates is used [53]. However, using only crop yield indicators, other factors that influence the quantity and quality of this service, such as properties of ecosystems and contribution of human management, are neglected [54].

The hierarchical tree designed for mapping provisioning service considers: a criterion for assessing soils potential, namely, LCC; and two criteria for assessing human inputs, namely, crop destination and type of activity. LCC expresses soil potential for agricultural use. Crop destination and type of activity describe the contribution of human management. The following types of activity are considered: non-professional, professional conventional, and professional organic. Crops, reclassified according to their main destination, are not evaluated for their productivity but based on a judgment of relative relevance among them. Crops for food are compared based on the following three types of value: economic, caloric, and health value. Crops for fibers and materials and crops for various destinations are compared based on economic and caloric values; crops for fodder for reared animals are compared only based on caloric value (all three types of value are given the same relevance).

### 2.4.2. Soil Quality

The formation and maintenance of soil structure and fertility through weathering and decomposition processes is a crucial service for plants growth and ecosystem functionality. This service is typically mapped using available data on soil properties and conditions [55].

The hierarchical tree designed for mapping soil quality service considers three criteria for assessing soils potential, namely, their physical, chemical, and hydraulic properties, and three criteria for assessing human inputs, namely, EFAs, crop type, and type of activity. The considered soil physical properties are texture, skeleton, and depth. The considered soil chemical properties are fertility, organic matter, and salinity. The considered soil hydraulic properties are available water capacity, water content, saturated hydraulic conductivity, and internal drainage. Crops, reclassified according to their type, and EFAs are evaluated for their ability to contribute to soil structure and fertility.

### 2.4.3. Erosion and Mass Movements Control

Soil protection from erosion and mass movements is an essential service both for agricultural and forestry productivity, and for the safety of settlements and infrastructures. Mapping of this service is often carried out by estimating land susceptibility to these processes, and vegetation contribution to land protection [56].

The hierarchical tree designed for mapping erosion and mass movements control service considers two criteria for assessing land potential to avoid or resist these processes, namely, soils physical properties and their vulnerability, and four criteria for assessing human inputs, namely, hydrogeological constraint, EFAs, length of crop cycle and type of activity. The considered soil physical properties are slope, hydrologic group, and texture. Soil vulnerability is assessed considering potential surface erosion, landslide hazard, and actual landslides. The hydrogeological constraint identifies areas protected pursuant to RD 3267/23. Crops, reclassified according to the length of their cycle, and EFAs are evaluated for their ability to contribute to soil protection.

### 2.4.4. Water Flows Regulation

The hydrological cycle and water flows regulation is a fundamental service to ensure stable and sufficient water availability for biotic communities and people, and to prevent extreme fluctuations, such as floods or droughts. The hydrological cycle regulation service is difficult to map because water flows are influenced by many interdependent processes, which ideally should all contribute to the storage of water and the reduction in surface runoff [57]. Soil is an essential component of the hydrological cycle, and its properties profoundly influence water flows regulation [58].

The hierarchical tree designed for mapping water flows regulation service considers four criteria for assessing soil potential to retain water and favor its infiltration—namely, soil moisture, soil hydraulic, and physical properties—and soil distance from surface waters, and three criteria for assessing human inputs, namely, EFAs, length of crop cycle, and type of activity. Soil moisture is assessed considering water shortage risk and the annual average precipitation for the period from 1995 to 2014. The distance from surface waters is assessed considering configuration of hydrographic network and hydraulic hazard. Soil hydraulic properties are assessed in the same way as the soil quality service. Soil physical properties are assessed in the same way as the erosion and mass movements control service. EFAs and crops (reclassified according to the length of their cycle) are evaluated for their ability to contribute to water retention and infiltration.

### 2.4.5. Maintenance of Habitats

Maintenance of habitats and of their biodiversity is the service that guarantees ecosystems functionality, and, therefore, influences the production of all other ESs, as well as the possibilities for future evolution. The mapping of this indispensable service is often carried out by estimating habitats suitability for one or more species, based on their distribution and on the distribution of the variables that control it [59].

The hierarchical tree designed for mapping maintenance of habitats service considers three criteria for assessing habitats potential, namely, their ecological value, their vulnerability, and the structure of the regional ecological network, and four criteria for assessing human inputs, namely, protected areas, EFAs, crop type, and type of activity. The vulnerability of habitats is assessed considering ecological sensitivity and anthropogenic pressure. The regional ecological network is evaluated according to its structural elements, i.e., different components of forest and agricultural ecosystems, and to its functional elements, i.e., components with high value or criticality. The protected areas comprise national and local parks and reserves. EFAs and crops (reclassified according to their type) are evaluated for their ability to contribute to habitat quality.

*2.5. Processing*

The proposed method exploits both computational capacities offered by raster format and data storage capacity offered by vector format in GIS. Figure 2 shows the flowchart of the process of ES mapping (left) and, as an example, the hierarchical tree for mapping the provisioning service (right).

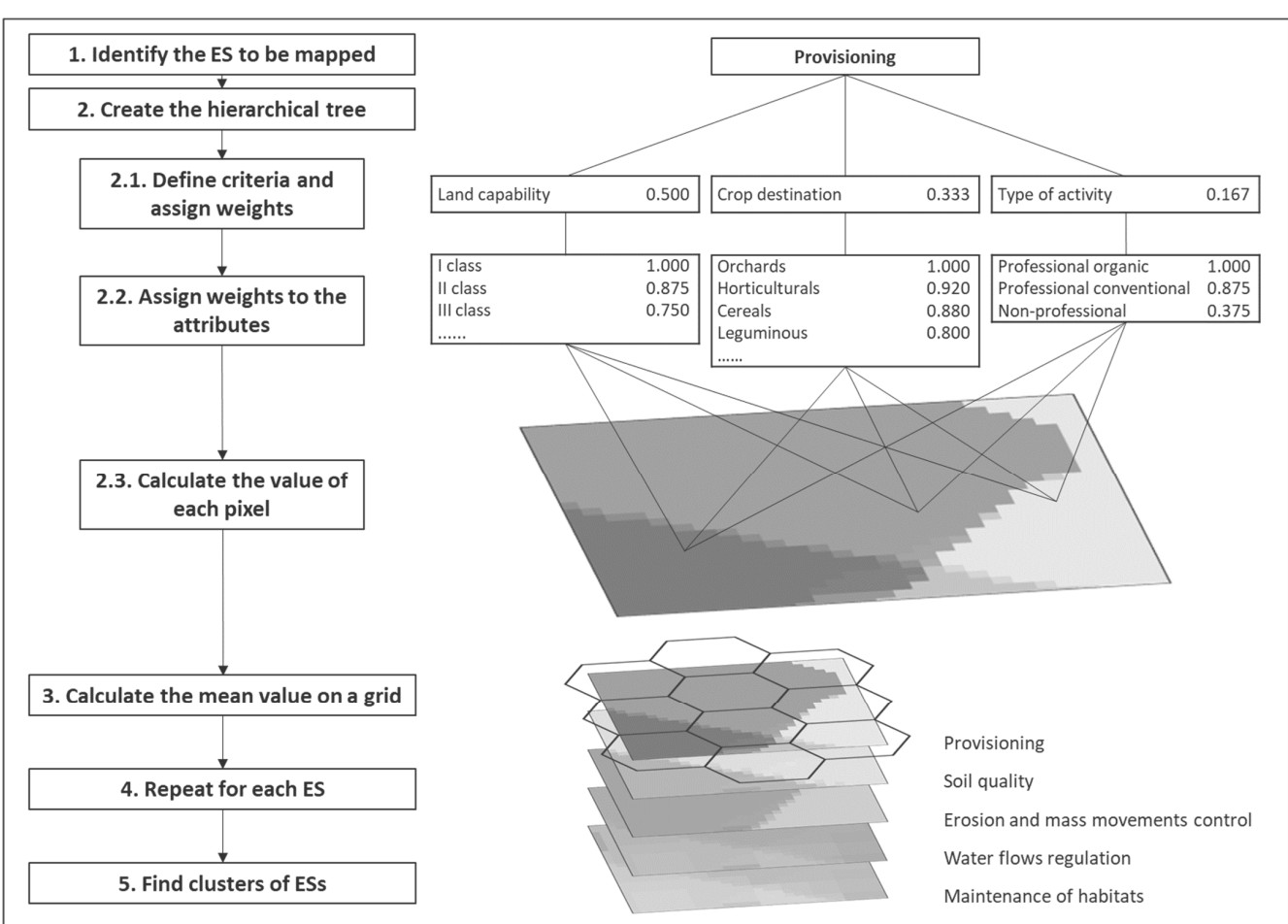

**Figure 2.** Flowchart of the process of ES mapping and example for the provisioning service.

Firstly, the hierarchical trees are defined, and weights are assigned to the criteria and attributes. Then, weights assigned to attributes are transferred to the spatial units (polygons) of the various maps used, for each criterion or sub-criterion of the five designed hierarchical trees.

Secondly, all maps are converted into raster format (resolution 5 × 5 m). The maps obtained from the overlay of the graphic cultivation plans (GCPs) on LULC maps for the years 2016/2020 are separately converted into raster format, and then the average

value of the weights taken by the pixels for the five years is calculated. Then, through map algebra operations, hierarchical trees are recomposed, multiplying maps by weights assigned to criteria and sub-criteria, and adding them at each level of the tree. Thus, five raster maps containing the general ordering of spatial units (pixels) according to their capacity to produce the ESs considered are obtained.

Thirdly, a vector grid of hexagonal polygons, with an area of approx. 1 ha, is created. This grid is much greater than the raster map's resolution, but compatible with the average dimensions of land covers. To each vector polygon of the grid, the average value of pixels contained within it is assigned, for each of the five raster maps of the ESs. These values are then normalized between 0 and 10 and used for characterizing the territory according to the capacity of different areas to produce the five ESs.

Furthermore, vector polygons collecting a value for each mapped ES allow for further statistical analysis. We performed a cluster analysis with the attribute-based clustering plug-in for QGis [60]. Cluster analysis is a set of multivariate data analysis techniques that aims to select and group homogeneous elements in a dataset [61]. Clustering techniques are based on relative measures of similarity between elements and can be hierarchical or non-hierarchical. In our case, we opted for the non-hierarchical technique known as k-means. This technique uses an unsupervised learning algorithm for identifying a fixed number of clusters within a dataset. With the attribute-based clustering plug-in, clusters are created grouping the elements of a vector layer based on the similarity of their attributes. In this way, the elements of the created hexagonal grid were grouped based on their values of ES supply. This analysis led to the identification of areas of the territory that are homogeneous in terms of capacity to provide a specific combination of the five mapped ESs. We tested the quality of results when using five, six, and seven clusters.

## 3. Results

### 3.1. Ecosystem Services Maps

The proposed method makes it possible to obtain a map of supply for each of the ESs considered. The five maps obtained subdivide the territory into homogeneous areas with a variable capacity to produce ESs, in relative terms, highlighting strengths and criticalities (Figure 3).

The provisioning map shows areas that have better land capability, and that in the last five years have been used for cultivating the most relevant crops, through professional agricultural activities. The provisioning service appears to be mainly located in the plain and in the first and lower range of hills, and especially in the central and eastern part of the plain. Higher hills and mountains have a lower capacity to supply this service.

The soil quality map shows areas with better physical, chemical, and hydraulic properties of soils, that have had less intensive use in agricultural activities, and that have been improved by means of EFAs. Soil quality service has an irregular distribution: the greatest supply capacity characterizes the mountains to the north of the area and partly also the mountains to the west and south; the lowest capacity is shown by some areas of the plain and in an area that crosses longitudinally the territory in the hills north of the plain.

The erosion and mass movements control map shows areas that are less vulnerable to these phenomena, and that during the last five years have been maintained under permanent or long-lasting vegetation and EFAs. The erosion and mass movements control service appears to be mainly located in the plain, due to its favorable topography. The hills and mountains have a variable capacity to supply this service, which is greater in the northeastern and in the southern part of the area.

The water flows regulation map shows the areas characterized by soils more conducive to water retention and infiltration, and that in the last five years have been used for less intensive agricultural activities, i.e., under permanent or long-lasting vegetation and EFAs. Water flows regulation service appears mainly located in the mountains to the north and to the south of the area. Lower hills and the plain have a lower capacity to supply this service.

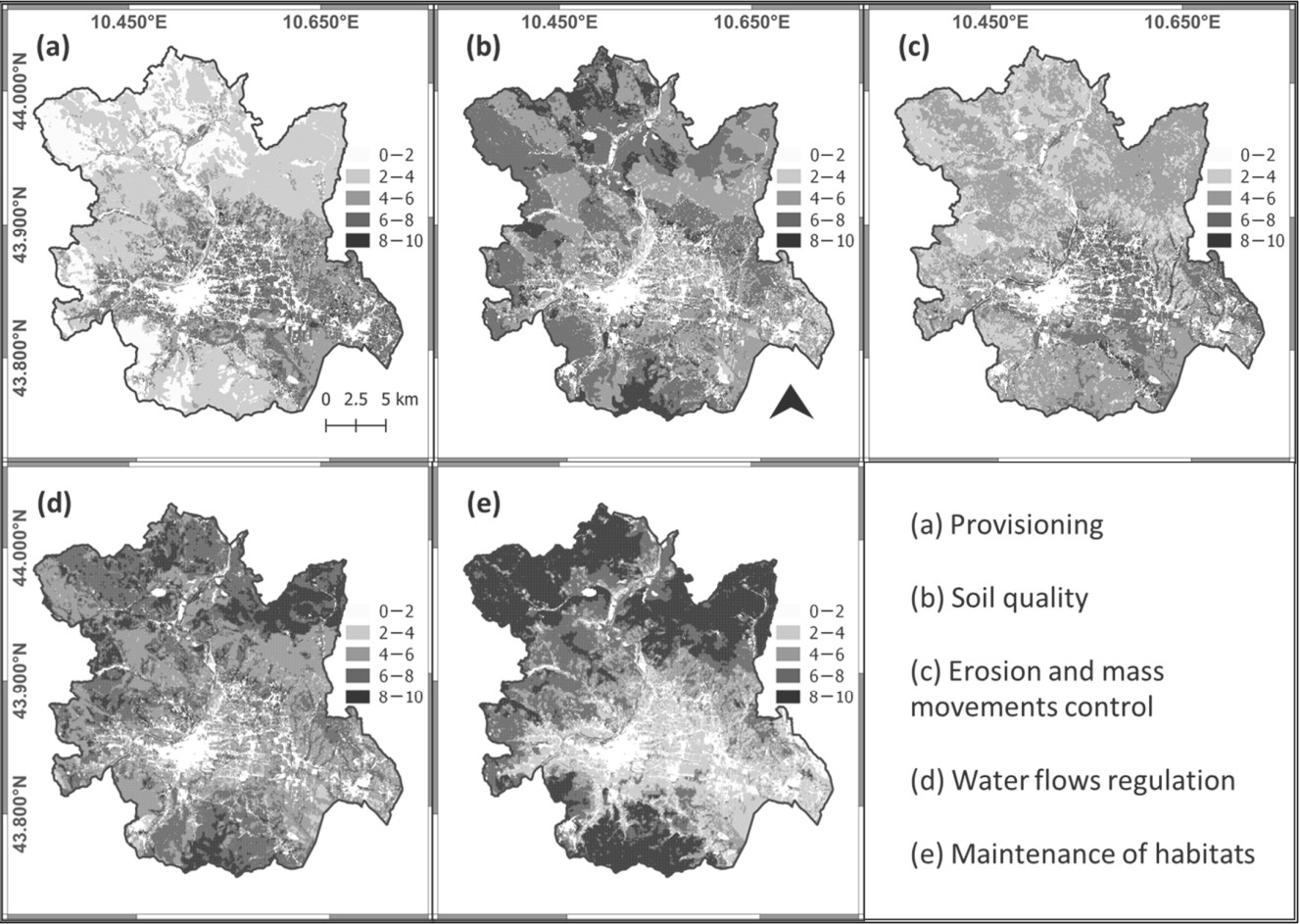

**Figure 3.** Ecosystem services maps.

The maintenance of habitats service map shows areas characterized by a higher ecological value and that, in the last five years, have had a less intensive use in agricultural activities and have been improved with the presence of EFAs. Habitat maintenance service is mainly located in the mountains, and to a lesser extent in the hills and plains.

*3.2. Ecosystem Services Bundles*

By using the obtained ES maps, cluster analysis makes it possible to identify areas that are more homogeneous in terms of capacity to supply different bundles of the five ESs considered. The territory is thus characterized according to six ES bundles that synthesize the capacity of different areas to supply different combinations of the five ESs mapped. The choice to use six bundles rests on the fact that this number, among those tested, is the one that best allows us to subdivide and characterize the territory. In fact, using five bundles, a relevant distinction between two of them disappears, while when using seven bundles, two bundles very similar to each other are created.

The ES bundles map shows the location of the six bundles obtained, and radar graphs express their relative supply capacity for the five ESs (Figure 4). This map highlights the existence of a characteristic bundle for most of the plain (B1), two bundles covering marginal and residual areas of the plain and the first lower range of hills (B2 and B3), while the remaining three bundles characterize the higher and more forested hills and mountains (B4, B5 and B6).

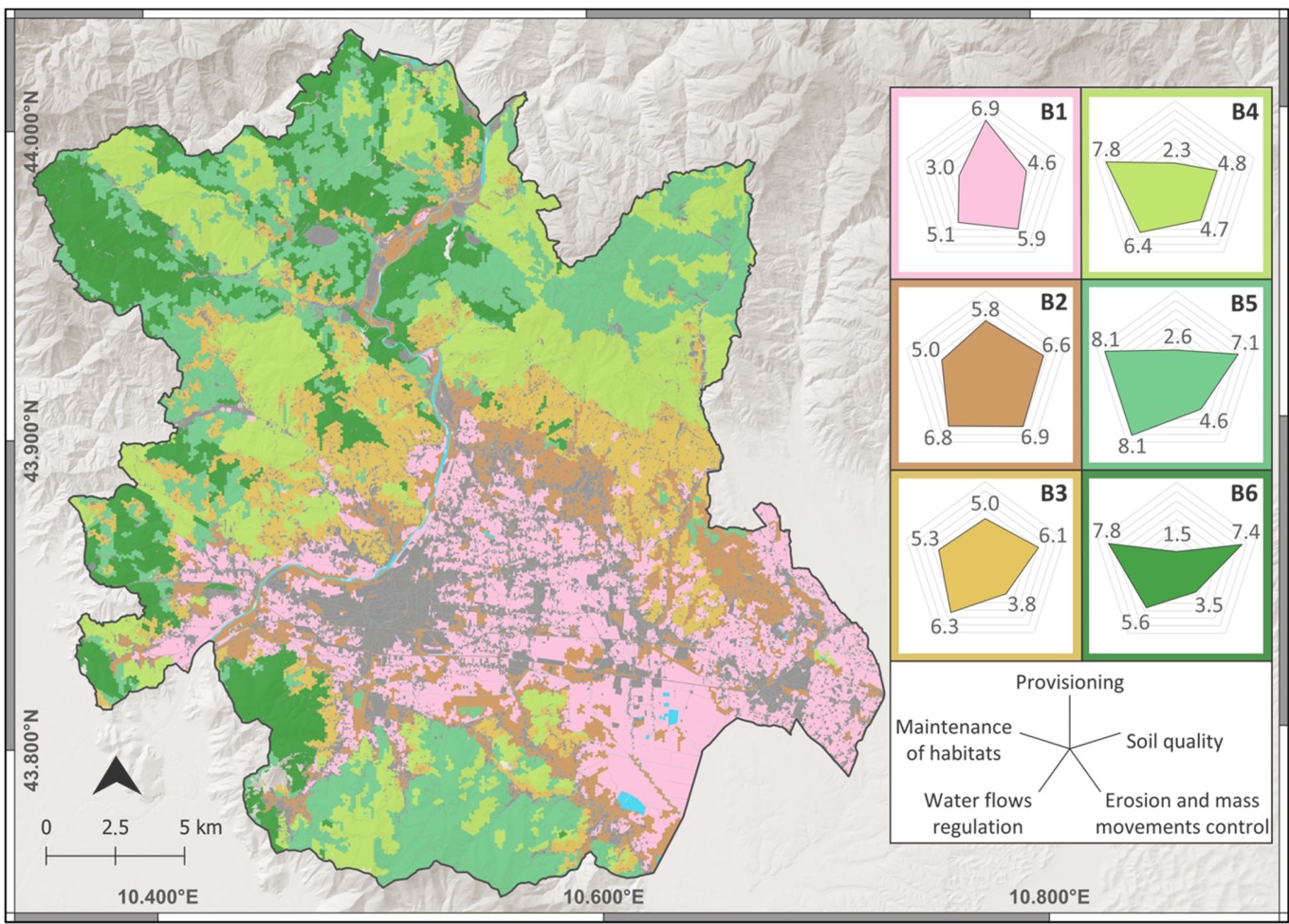

**Figure 4.** Ecosystem services bundles.

Bundle B1 amounts to 19.4% of the area's total surface and almost entirely covers the plain. This area is characterized by a high capacity for provisioning service (6.9), due to the presence of many agricultural activities, and by a good capacity for the erosion and mass movements control service (5.9), due to its topography. However, it is characterized by a lower supply capacity for the other regulating services considered: soil quality (4.6), water flows regulation (5.1), and above all, markedly, maintenance of habitats (3.0).

Bundle B2 amounts to 10.3% of the area total surface and collects marginal and residual areas of the plain and the foothills, partly covered by small tree rows, groves, and buffer strips, and partly subject to less intensive agriculture or abandonment. Compared to the area included in bundle B1, these areas are characterized by a lower capacity for provisioning service (5.8), but by a higher capacity for the maintenance of habitats service (5.0) and by a good capacity for the other regulating services: soil quality (6.6), erosion and mass movements control (6.9), and water flows regulation (6.8). These areas are, therefore, to be considered those with the most balanced supply of the five ESs mapped.

Bundle B3 amounts to 12.6% of the area total surface and covers the first lower hills, which often host typical permanent crops such as olive groves and vineyards. Like the areas included in bundle B2, these areas show a rather balanced supply of the five ESs mapped. Their capacity for soil quality service (6.1), water flows regulation service (6.3), and maintenance of habitats service (5.3), is very close to that of the areas in bundle B2, while their capacity for provisioning service (5.0) is slightly lower, and their capacity for erosion and mass movements control service (3.8) much lower, due to increasing slope.

Bundle B4 amounts to 21.2% of the area total surface and, together with bundle B5 (21.6% of the area total surface), represents the most widespread bundle in the area. It

includes areas of hills and mountains that show low capacity for provisioning service (2.3), for erosion and mass movements control service (4.7), and for soil quality service (4.8). However, these areas have good capacity for water flows regulation service (6.4), and high capacity for maintenance of habitat service (7.8).

Bundle B5 includes areas of hills and mountains characterized by the highest capacity for water flows regulation service (8.1), and for the maintenance of habitats service (8.1). Like areas included in other bundles describing hills and mountains mostly covered by forest (B4 and B6), these areas show a low capacity for provisioning service (2.6), and for erosion and mass movement control service (4.6). Unlike the areas included in bundle B4, these areas have high capacity for soil quality service (7.1).

Bundle B6 amounts to 14.8% of the area total surface and collects areas of hills and mountains characterized by a very low capacity for provisioning service (1.5), and erosion and mass movement control service (3.5). These areas also show low capacity for water flows regulation service (5.6), but high capacity for maintenance of habitats service (7.8), and very high capacity for soil quality service (7.4).

## 4. Discussion

This paper proposes a method for mapping the supply of five ESs produced by agricultural and forest areas applied to the Landscape Area of Lucchesia, in the Tuscany region (one of the Landscape Areas defined in the regional PIT-PPR), but potentially replicable in other contexts. The method combines an LULC map and other environmental, socio-economic, and geographical data to classify the areas of the territory according to their relative capacity to produce ESs. It considers both ecosystems potential and human inputs, as determinants to ES supply, and, therefore, it can lead to a more comprehensive and specific characterization of a territory than those based only on an LULC map or a single indicator. The use of the AHP technique, integrated with GIS, allows us to easily organize and combine both qualitative and quantitative spatial information, and to map ES supply through both indicators of soil and ecosystem properties and conditions, and indicators of human contributions in recent (2016/2020) land management. The results obtained by using this technique strongly depend on chosen hierarchical trees, and above all on weights assigned to the elements of the various levels of the trees. The design of the trees is influenced by data availability, and is the most rigid part of the method, while the assignment of weights requires data interpretation and is the most adjustable and questionable part of the method.

In this first test of the method, the weights were assigned to criteria and attributes by the authors. However, the method is designed to encourage discussion between experts and stakeholders with the aim of improving the latter's awareness of the importance of the territory in providing ESs. Involving stakeholders in the definition of the weights makes it possible to achieve two important objectives: to bring the parties together towards a shared assessment of the ESs, and to develop scenario analyses ("what would happen if . . . ") useful for defining strategies for sustainable territorial development. The results show that, even if further refinement and testing is required, the proposed method has the potential for innovating the information and decision support tools that require ES mapping.

A key issue in ES mapping is the choice of appropriate indicators and data. From this point of view, the Tuscany information system, integrated to a small extent with national sources, provides a rather good quantity and quality of data. The Regional Pedological Database completes the LULC map with information on soil properties and conditions, allowing us to assess the contribution of this valuable resource to ES supply. Unfortunately, data on Tuscan forest areas are poorly updated and lack details. Consequently, ES assessment could be improved if there was better information on forests' structure and condition. Vice versa, the ARTEA Database on agricultural areas is more detailed and updated annually, and it provides information on cultivated species and the cultivation method (conventional or organic), even if more precise information on the actual management practices of agricultural areas would improve ES assessment. However, the overlay

of ARTEA GCPs on LULC maps allows us to differentiate areas that have been managed by professional farms or by hobby farms during the last five years and deepen our knowledge of cultivation activities in the former.

The resulting ES maps subdivide the analyzed territory into areas with different capacities of ES supply and highlight their strengths and criticalities, thus providing useful indications to design territorial strategies and political and governance measures at many levels and in many sectors. Maps offer a general overview of ES supply distribution across the territory of the eight municipalities included in the landscape area. In addition, if focusing on specific areas, they can give a more detailed description.

Furthermore, cluster analysis allows us to identify homogeneous areas in terms of capacity to supply different ES bundles, and it helps to focus strategies and measures not just on some isolated ESs, but on a combination of them. Many researchers have identified ES bundles with a variety of assessment methods, both at national scale, with low resolution grids [62], at regional scale, with administrative divisions used as spatial units [63], and at city scale [64]. The method proposed in this paper, by defining bundles of the ESs considered with a grid of resolution 1 ha for the analyzed area, a territory already described as homogeneous in the main regional planning tool, can deepen the knowledge of its capacity to produce different ESs in different locations, and of the relationships between them.

The obtained ES bundles show the trade-off between the provisioning service and regulating services already found in other contexts [65,66]. In fact, the areas with high capacity for the provisioning of food and materials show a correspondingly low capacity for regulating services, probably due to the presence of intensive agriculture. Vice versa, the areas with high capacity for regulating services, such as the maintenance of habitats, usually have a low capacity for provisioning service. In particular, the characteristic bundle of the plain (B1), where there are both arable land and heterogeneous agricultural areas, has a high capacity for provisioning service, but low capacity for regulating services, especially for maintenance of habitats. Instead, bundles that cover marginal areas of the plain and first range of lower hills (B2 and B3), where arable land, heterogeneous areas, permanent crops, and natural and semi-natural areas are mixed, have the most balanced capacity for the different ESs. On the contrary, bundles that characterize higher hills and mountains (B4, B5 and B6), where natural and semi-natural areas, especially forests, dominate, have in common a low capacity for provisioning service, but high capacity for regulating services, albeit with some differences.

While ES bundles can give first indications on the relationships among ESs, further research is needed to understand the factors that determine their association, to provide more specific and useful information to planners, administrators and farmers. Mapping ES bundles is a fundamental issue for the application of ES research in decision-making processes, for instance, to include them in planning activities [67], or to promote the introduction of systems of payment for ES (PES) [68]. The proposed method offers a first estimate of the distribution of bundles of the five ESs considered within the analyzed area, and potentially can allow stakeholders to assess ES availability and variability, to decide where and how to intervene with more in-depth investigations.

In our case study (the Landscape Area of Lucchesia), for example, the results suggest two directions to follow in planning:

- In the plain, there is high capacity for the provisioning service but, at the same time, low capacity for regulating services. Therefore, to maintain the former, policies to limit urban development, and to improve the latter, policies to encourage organic farming are recommended.
- In hills and mountains, there is high capacity for regulating services which, among other things, are useful to the population concentrated in the plain. Therefore, since these areas are subject to increasing abandonment of agricultural and forestry activities, policies would be needed to ensure adequate incomes for farmers.

The planning and management of multifunctional forms of agriculture and forestry can help to achieve these objectives, but the implementation of effective strategies often encounters difficulties due to the long adaptation times required by farmers, which tend to be reluctant to change. In this respect, it is necessary to propose innovative forms of governance such as, for example, public–private partnerships or PES agreements. To be developed, these forms of governance require appropriate information and knowledge, and tools such as the one proposed in this paper.

The proposed method for ES mapping demonstrates very interesting results, but also has room for further improvement. Firstly, with reference to the CICES classification, it is necessary to widen the set of considered ESs, e.g., including the cultural services of agricultural and forest areas, which offer very important benefits requested by people. Secondly, another point to be explored is how to evaluate ESs by separately analyzing, on one hand, an ecosystem's potential (intrinsic characteristics) and, on the other, the role played on it by human activities (management features). This approach would make it possible to verify whether human management is coherent with the vocation of the ecosystems, as well as the effects of different management practices. Furthermore, it could be interesting to map ES bundles on other scales—both larger, for example, a single municipality or a specific area, and smaller, for example, a group of Landscape Areas of the PIT-PPR, or a Landscape System [69]—to identify different configurations of the relationships between services. An important object of study could also be the effect of spatial resolution on ES assessment and bundling, with the aim to identify the best compromise between effectiveness and costs of the analysis. Finally, if we set the goal of defining territorial balances between ES supply and demand, it is necessary to open a further front of research on data and methods for evaluating ES demand [70], being aware that demand is often located at distant places from the areas where ES supply is produced [71].

## 5. Conclusions

In many regions of Europe, urbanization processes in the plains and the abandonment of cultivated land in hilly and mountain areas are reducing food production and causing significant impacts on key environmental resources such as soil, water, and biodiversity, both at local and global scale. In the future, food production and human well-being will increasingly depend on the ability to conserve and manage these fundamental resources in a sustainable way, supporting and enhancing the ability of agricultural ecosystems to provide a balanced mix of ESs. Territorial planning needs to promote adequate balances between supply and demand of resources and ESs at local level, through better integration and reciprocity between urban, peri-urban, and rural areas. Agricultural and forest ecosystems in rural and peri-urban areas produce fundamental ESs for people's quality of life that must be recognized, evaluated, protected, and valued through innovative governance models.

The method proposed in this paper represents an attempt to integrate an LULC map with other geographical, environmental, and socio-economic data, for mapping the supply of five ESs selected from the CICES classification. It also provides an estimate of the relationships between the mapped ESs by identifying areas homogeneous in terms of capacity to supply different bundles of them. Cluster analysis makes possible an approach based on ES bundles that is much more effective in characterizing territorial specificities than a map of individual ESs, because it provides an integrated view of the mix of ESs produced, and highlights trade-offs between the ESs mapped. Indeed, within the case-study area, there are areas characterized by a very efficient capacity to supply one or two ESs but to the detriment of other ESs, and areas where the capacity to supply the five ESs is more balanced, and consequently that may be connoted as multifunctional areas. This approach provides us with a better understanding and knowledge of a territory, and it can be paramount if used in a DSS for identifying the strengths and weaknesses of the various areas of a territory as regards their capacity to provide ESs. This information is

essential to design adequate intervention strategies and actions for guaranteeing ecosystem sustainability and resilience.

**Supplementary Materials:** The following supporting information can be downloaded at: https://www.mdpi.com/article/10.3390/land12061123/s1, Table S1: Hierarchical trees and weights; Table S2: Attributes and weights.

**Author Contributions:** Conceptualization, M.R., T.T. and F.M.; methodology, M.R., T.T. and F.M.; data curation, M.R. and T.T.; software, T.T.; visualization, T.T.; writing—original draft preparation, T.T.; writing—review and editing, M.R., T.T. and M.A. All authors have read and agreed to the published version of the manuscript.

**Funding:** This research received no external funding.

**Data Availability Statement:** Used data can be found in the Tuscany Region webgis: https://www502.regione.toscana.it/geoscopio/cartoteca.html (last accessed on 23 March 2023) and https://dati.toscana.it/ (last accessed on 23 March 2023); the National webgis: http://www.pcn.minambiente.it/mattm/servizi-ogc/ (last accessed on 23 March 2023); and the ISPRA website: https://www.isprambiente.gov.it/it/banche-dati (last accessed on 23 March 2023).

**Acknowledgments:** We thank Giovanni Zanchetta, Monica Bini, Marco Luppichini, and Federico Bellesi for their valuable support to this research.

**Conflicts of Interest:** The authors declare no conflict of interest.

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
