# Peer review of "Mapping Ecosystem Services Bundles for Spatial Planning with the AHP Technique: A Case Study in Tuscany (Italy)"

_land, doi:10.3390/land12061123_

Round 1

Reviewer 1 Report

Comments to the author:

The paper entitled “Mapping Ecosystem Services Bundles for Spatial Planning by Combining Territorial Data with the AHP. A Case-Study in Tuscany (Italy).” proposes a methodology for mapping and bundling ES supply by agricultural and forests areas, combining territorial data with the AHP technique.  

The topic is challenging, but it is undoubtedly worthy of investigation. It falls within the aim of the journal, it could be of interest for the readers, and the overall merit of the paper is good, if it must be improved.

The paper is well structured; the writing is correct; the English language is correct.

The title provides a good contextualization of the work but it can be improved. 

The abstract is clear; nevertheless some revisions are necessary.

The introduction provides sufficient background for the research, some paragraphs have to be revised, rephrased and clarified.

The methodology is quite robust and supported by an appropriate bibliography; some aspects are addressed too concisely or not at all. They should be expressed more clearly and extensively. Different information and charts (work process, criteria tree, etc.) are missing.

Results and discussion are consequent with the main thesis addressed by the study. Overall, the paper increases the awareness on the importance of integrated approaches in the landscape governance, as measure to enforce the relationship between natural and social systems and instances.

The fluent reading of the paper is exacerbated by the excessive use of the inserted abbreviations, resulting very hard to follow. I suggest revising all the description, just enlarging the concept already present, in a way to facilitate the reader and the comprehension.

Specific comments:

·         Title: “Ecosystem Services Bundles …”: Do not use capital letters

·         Title: “AHP”: I suggest to add “technique or method, etc.

·         Title: A Case-Study in Tuscany (Italy).”: without point.

·         Lines 11-13: The concept is unclear, please rephrase.

·         Line 28: “agricultural area expansion”. Do you refer to global scale? For Italy and Europe, somewhere in the paper, you refer to a reduction in agricultural land. Please, explain further.

·         Lines 32-33: “North … South”: Do you refer to a global scale? Please, rephrase.

·         Lines 35-36: The concept is unclear, please rephrase.  

·         Lines 37-40: The concept is unclear treated in a generic and non-contextual way, please rephrase.  

·         Line 56: “food sovereignty”. The concept is unclear, please rephrase.  

·         Lines 62-64: Do you refer to different point of view or approaches? Please rephrase.

·         Lines 68: “or in other region of the world”. The concept is unclear, please rephrase.  

·         Line 95: “characterized by non-rivalry and non-excludability”. The concept is unclear, please rephrase.  

·         Lines 140-144: Are you sure?

·         Lines 148-151: The concept is unclear, please rephrase.  

·         Lines 160-163: “… has a good international reputation due to its landscapes [44] and its high-quality agricultural products, resources that 161 must be protected through the maintenance and enhancement of rural areas, with effective measures and adequate information tools [45]”. Please rephrase.

·         Lines 171-173. The concept is unclear, please rephrase and explain more extensively.

·         Lines 182-184. The concept is repeated, please delete it.

·         Lines 203-205. The concept is unclear, please rephrase.  

·         Line 215. A chart entailing the work process and a description on the steps in methods would be needed.

·         Lines 228-230: Why do you use only specific classes belonging to the second and third CLC classification level? You have to explain the reasons and the criteria used.

·         Lines 232-247. Excessive use of the inserted abbreviations.

·         Line 242: (Section 2.4). The section refers to the Ecosystem Services and not to the reclassification by homogeneous groups.

·         Lines 286-292. The concept is unclear, please rephrase.

·         Lines 297-314: The concept is unclear, please rephrase.

·         Lines 320-322: The Hierarchical trees for each ES and weights assigned to criteria are missing.

·         Lines 342-351. What is the criterion for selected ecosystem services? Please explain extensively.

·         Lines 354. A graph or table is needed with a summary of the ecosystem services selected, the analysis methodologies adopted and the criteria used.

·         354-453: Established methodologies and dedicated software already exist, for evaluating each ecosystem service analysed. Why did you not refer to them?

·         Line453: A description, concerning the methodology used for the cluster analysis of ecosystem services, is completely missing.

·         Lines 460-480: Are these results or the weights given?

·         Line 541: It is missing any reference to the demand for ecosystem services. Could this information be integrated?

·         Lines 562-565: What stakeholders? How was uncertainty handled in the process?

·         Lines 599-600: The concept is unclear, please rephrase.

·         Lines 620-629: You need to describe the phenomenon, in the study-area section perhaps, in order to discuss it here.

·         Lines 630-632: The concept is unclear, please rephrase.

No comments

Author Response

Reviewer 1 - Comments and Suggestions for Authors

Comments to the author:

The paper entitled “Mapping Ecosystem Services Bundles for Spatial Planning by Combining Territorial Data with the AHP. A Case-Study in Tuscany (Italy).” proposes a methodology for mapping and bundling ES supply by agricultural and forests areas, combining territorial data with the AHP technique.  

The topic is challenging, but it is undoubtedly worthy of investigation. It falls within the aim of the journal, it could be of interest for the readers, and the overall merit of the paper is good, if it must be improved.

The paper is well structured; the writing is correct; the English language is correct.

Thank you very much.

The title provides a good contextualization of the work but it can be improved. 

The title has been shortened.

The abstract is clear; nevertheless some revisions are necessary.

The abstract has been revised.

The introduction provides sufficient background for the research, some paragraphs have to be revised, rephrased and clarified.

Some paragraphs have been removed, others revised, according to the specific comments.

The methodology is quite robust and supported by an appropriate bibliography; some aspects are addressed too concisely or not at all. They should be expressed more clearly and extensively. Different information and charts (work process, criteria tree, etc.) are missing.

The methodology has been explained more in some parts of the manuscript (2.2-2.5). Two tables and a flow chart with an example have been included.

Results and discussion are consequent with the main thesis addressed by the study. Overall, the paper increases the awareness on the importance of integrated approaches in the landscape governance, as measure to enforce the relationship between natural and social systems and instances.

The fluent reading of the paper is exacerbated by the excessive use of the inserted abbreviations, resulting very hard to follow. I suggest revising all the description, just enlarging the concept already present, in a way to facilitate the reader and the comprehension.

We have removed some abbreviations, according to the specific comments.

Specific comments:

Title: “Ecosystem Services Bundles …”: Do not use capital letters Done

Title: “AHP”: I suggest to add “technique or method, etc. Done

Title: A Case-Study in Tuscany (Italy).”: without point. Done

Lines 11-13: The concept is unclear, please rephrase. Abstract has been revised.

Line 28: “agricultural area expansion”. Do you refer to global scale? For Italy and Europe, somewhere in the paper, you refer to a reduction in agricultural land. Please, explain further. REMOVED

Lines 32-33: “North … South”: Do you refer to a global scale? Please, rephrase. REMOVED

Lines 35-36: The concept is unclear, please rephrase.  REPHRASED

Lines 37-40: The concept is unclear treated in a generic and non-contextual way, please rephrase.  REMOVED

Line 56: “food sovereignty”. The concept is unclear, please rephrase.  REPHRASED

Lines 62-64: Do you refer to different point of view or approaches? Please rephrase. REPHRASED

Lines 68: “or in other region of the world”. The concept is unclear, please rephrase.  REPHRASED

Line 95: “characterized by non-rivalry and non-excludability”. The concept is unclear, please rephrase.  REMOVED AND REPHRASED

Lines 140-144: Are you sure? REPHRASED

Lines 148-151: The concept is unclear, please rephrase.  REPHRASED

Lines 160-163: “… has a good international reputation due to its landscapes [44] and its high-quality agricultural products, resources that 161 must be protected through the maintenance and enhancement of rural areas, with effective measures and adequate information tools [45]”. Please rephrase. REPHRASED

Lines 171-173. The concept is unclear, please rephrase and explain more extensively. DONE

Lines 182-184. The concept is repeated, please delete it. DONE

Lines 203-205. The concept is unclear, please rephrase.  

Line 215. A chart entailing the work process and a description on the steps in methods would be needed. DONE IN PAR. 2.5

Lines 228-230: Why do you use only specific classes belonging to the second and third CLC classification level? You have to explain the reasons and the criteria used. DONE

Lines 232-247. Excessive use of the inserted abbreviations. We have removed some abbreviations.

Line 242: (Section 2.4). The section refers to the Ecosystem Services and not to the reclassification by homogeneous groups. We have written the reclassifications we have made and included a table (1)

Lines 286-292. The concept is unclear, please rephrase.

Lines 297-314: The concept is unclear, please rephrase. REPHRASED

Lines 320-322: The Hierarchical trees for each ES and weights assigned to criteria are missing. A table (2) has been included.

Lines 342-351. What is the criterion for selected ecosystem services? Please explain extensively. EXPLAINED

Lines 354. A graph or table is needed with a summary of the ecosystem services selected, the analysis methodologies adopted and the criteria used. A table (2) and a flow chart (Figure 2) have been included.

354-453: Established methodologies and dedicated software already exist, for evaluating each ecosystem service analysed. Why did you not refer to them? We have referred to software in the introduction. There are also brief references before the description of each hierarchical tree.

Line453: A description, concerning the methodology used for the cluster analysis of ecosystem services, is completely missing. Done

Lines 460-480: Are these results or the weights given? These are the results.

Line 541: It is missing any reference to the demand for ecosystem services. Could this information be integrated? The paper focused only on the supply of the mapped ecosystem services. Regarding the assessment of the demand, it would be necessary to investigate where the population is concentrated, production activities and consumption of goods and services, etc., but this was not the primary objective of the research.

Lines 562-565: What stakeholders? How was uncertainty handled in the process? REPHRASED

Lines 599-600: The concept is unclear, please rephrase. REPHRASED

Lines 620-629: You need to describe the phenomenon, in the study-area section perhaps, in order to discuss it here. Done in par 2.1.

Lines 630-632: The concept is unclear, please rephrase. REPHRASED

Reviewer 2 Report

Overall an excellent and interesting paper, one or two minor changes and some suggestions to improve the understanding.

1. Editing errors - line 296 - reference to Rovai and Andreoli (2018) in text should this be referenced with a number rather than your names? or is this intentional because it is by yourselves?

2. line 675: delete "it"

3. It would make a clearer understanding of the structuring of the Ecosystem service groups to provide a comparative table with the criteria/attributes that are considered for each of the 5 ES

4. For readers unfamiliar with the AHP techniques it would be useful to give a worked example diagram of how the four levels, Goals, criteria and sub-criteria, attributes and alternatives are used to assign the weights in the hierarchy. or are these worked out in supplementary material to the paper?

5. Similarly a worked example of the bundling process as a diagram would also be helpful, especially as you say that in this case study the weights and bundles are based upon your own personal judgement, or are these already in the supplementary material?

6. To what extent do the zones of bundles correspond to other landuse classifications and planning zones, or are the bundles similar in extent. It would be useful to make the comparison in the discussion to show that this approach comes up with very different zones, or rather provide added value to the planning zones with the extra analysis of the contributions of different ecosystem services. From this you could then discuss how this analysis can contribute practically to the PIT-PPR decisions. This would round off the paper to show the usefulness of the approach. 

Author Response

Reviewer 2 - Comments and Suggestions for Authors

Overall an excellent and interesting paper, one or two minor changes and some suggestions to improve the understanding.

Thank you very much.

  1. Editing errors - line 296 - reference to Rovai and Andreoli (2018) in text should this be referenced with a number rather than your names? or is this intentional because it is by yourselves?

It was an error.

  1. line 675: delete "it"

Done

  1. It would make a clearer understanding of the structuring of the Ecosystem service groups to provide a comparative table with the criteria/attributes that are considered for each of the 5 ES

A table (2) has been included.

  1. For readers unfamiliar with the AHP techniques it would be useful to give a worked example diagram of how the four levels, Goals, criteria and sub-criteria, attributes and alternatives are used to assign the weights in the hierarchy. or are these worked out in supplementary material to the paper?

A flow chart with an example has been included in the manuscript (Figure 2). Complete hierarchical trees are in supplementary material, tables S1 e S2.

  1. Similarly a worked example of the bundling process as a diagram would also be helpful, especially as you say that in this case study the weights and bundles are based upon your own personal judgement, or are these already in the supplementary material?

A description concerning the methodology used for the cluster analysis has been included in par. 2.5.

  1. To what extent do the zones of bundles correspond to other landuse classifications and planning zones, or are the bundles similar in extent. It would be useful to make the comparison in the discussion to show that this approach comes up with very different zones, or rather provide added value to the planning zones with the extra analysis of the contributions of different ecosystem services. From this you could then discuss how this analysis can contribute practically to the PIT-PPR decisions. This would round off the paper to show the usefulness of the approach. 

See lines 642-651 and 661-670 of the revised manuscript.

Reviewer 3 Report

General comments:
Mapping ecosystem services bundles can provide scientific basis for establishing reasonable territorial spatial planning. While the management and regulation of ESs are complex and require scientifically sound and widely understandable policies and governance models, based on detailed assessment methods. This manuscript proposes a method for mapping and bundling the supply of five ESs produced in agricultural and forest areas, based on the processing of open-source territorial data through the AHP, and tailored for Tuscany Region. However, the innovation of the study is not outstanding. What's more, there is still a certain gap between theoretical research and practical application. The manuscript needs to further explain and improve clearly and comprehensively. Some details concern about the manuscript are given below.
1. There are too many paragraphs and the content is too scattered in this manuscript. It is suggested to simplify and reorganize the relevant content, especially the sections of introduction, materials and methods and discussion, or even one sentence as a paragraph.
2. The research background in the introduction section is too long, so it should be concise and comprehensive, highlighting the key points, and directly pointing out the study questions and contents. The introduction should play the role of connecting the preceding and the following, and finally should lead to the following key contents and clarify the research objectives of this manuscript.
3. In addition, the introduction section should summarize the research progress, analyze the relationship between this manuscript and the current hot issues, and summarize what work has been carried out and what contributions have been made by previous scholars in this field. Through summarizing, drawing experience, and analyzing the unsolved problems, the scientific problems of this manuscript are proposed. So, the introduction section needed to be rewritten.
4. The subsection of 2.3 and 2.4 should belong to the Method section, and this section should not only be a simple description of methods, but also a detailed introduction to the selection of indicators, parameter settings and so on in this manuscript. For example, it should clearly explain the indicators involved in calculating each ecosystem service and its weight, and explain the basis for determining bundles number in cluster analysis. At the same time, it is necessary to add a technical flow chart to clearly and intuitively demonstrate the research processes, research contents, corresponding data and methods.
5. It is not clear which year's LULC data was used for ESs mapping in Figure 2? The manuscript mentions the LULC maps of the years 2016 and 2019 with the codes of the agricultural and forest areas were used for ESs mapping (line223~224), but there was only one result.
6. At present, many ESs integrated models have been used for ESs evaluation. The advantages of ecosystem service evaluation methods proposed in this manuscript should be explained.
7. Regional spatial planning should not only focus on agricultural and forest areas, but
should involve the whole region. At the end of the study, specific optimization
suggestions should be put forward for each bundle type classified in this manuscript.
8. This manuscript only classified the types of ESs bundles, and does not analyze the
formation mechanism and influence mechanism of each bundle. Carrying out relevant
study will provide more feasible optimization suggestions for territorial spatial planning.

 Minor editing of English language required.

Author Response

Reviewer 3 - Comments and Suggestions for Authors

General comments:

Mapping ecosystem services bundles can provide scientific basis for establishing reasonable territorial spatial planning. While the management and regulation of ESs are complex and require scientifically sound and widely understandable policies and governance models, based on detailed assessment methods. This manuscript proposes a method for mapping and bundling the supply of five ESs produced in agricultural and forest areas, based on the processing of open-source territorial data through the AHP, and tailored for Tuscany Region. However, the innovation of the study is not outstanding. What's more, there is still a certain gap between theoretical research and practical application. The manuscript needs to further explain and improve clearly and comprehensively. Some details concern about the manuscript are given below.

Thank you very much.

  1. There are too many paragraphs and the content is too scattered in this manuscript. It is suggested to simplify and reorganize the relevant content, especially the sections of introduction, materials and methods and discussion, or even one sentence as a paragraph.

DONE

  1. The research background in the introduction section is too long, so it should be concise and comprehensive, highlighting the key points, and directly pointing out the study questions and contents. The introduction should play the role of connecting the preceding and the following, and finally should lead to the following key contents and clarify the research objectives of this manuscript.

The introduction has been shortened and revised.

  1. In addition, the introduction section should summarize the research progress, analyze the relationship between this manuscript and the current hot issues, and summarize what work has been carried out and what contributions have been made by previous scholars in this field. Through summarizing, drawing experience, and analyzing the unsolved problems, the scientific problems of this manuscript are proposed. So, the introduction section needed to be rewritten.

The introduction has been shortened and revised.

  1. The subsection of 2.3 and 2.4 should belong to the Method section, and this section should not only be a simple description of methods, but also a detailed introduction to the selection of indicators, parameter settings and so on in this manuscript. For example, it should clearly explain the indicators involved in calculating each ecosystem service and its weight, and explain the basis for determining bundles number in cluster analysis. At the same time, it is necessary to add a technical flow chart to clearly and intuitively demonstrate the research processes, research contents, corresponding data and methods.

Section 2 (materials and methods) now contains: 2.1. study area; 2.2. description of the AHP method; 2.3. data; 2.4. application of the method (hierarchical trees); 2.5. processing in gis. A flow chart with an example has been included in the manuscript (Figure 2). Complete hierarchical trees are in supplementary material, tables S1 e S2.

  1. It is not clear which year's LULC data was used for ESs mapping in Figure 2? The manuscript mentions the LULC maps of the years 2016 and 2019 with the codes of the agricultural and forest areas were used for ESs mapping (line223~224), but there was only one result.

LULC map of the year 2019 was used in Figure 1. In Figure 3 (former Figure 2) the contribution of land use (human management) to the supply of ESs was calculated as the average value for the five years. See lines 269-271 of the revised manuscript (“The GCPs of the years 2016/2017 were overlaid on the selected elements of the 2016 LULC map, while the GCPs of the years 2018/2020 on those of the 2019 LULC map”). And lines 661-670 of the revised manuscript (“The maps obtained from the overlay of the Graphic Cultivation Plans (GCPs) on LULC maps for the years 2016/2020 are separately converted into raster format, and then the average value of the weights taken by the pixels for the five years is calculated”).

  1. At present, many ESs integrated models have been used for ESs evaluation. The advantages of ecosystem service evaluation methods proposed in this manuscript should be explained.

See lines 92-109 of the revised manuscript.

  1. Regional spatial planning should not only focus on agricultural and forest areas but should involve the whole region. At the end of the study, specific optimization suggestions should be put forward for each bundle type classified in this manuscript.

We fully agree on this point, but in this specific study we were interested in investigating a method to suggest to planners to enhance the role of the agroforestry territory in providing useful ESs for citizens. In Italy, in fact, this is the part of the territory most subjected to a reduction / erosion due to urbanization processes or due to the abandonment of agricultural activity.

  1. This manuscript only classified the types of ESs bundles and does not analyze the formation mechanism and influence mechanism of each bundle. Carrying out relevant study will provide more feasible optimization suggestions for territorial spatial planning.

Yes, we agree with this observation and for this reason we wanted to verify if the cluster analysis could be validly used. As explained in the discussion of the results, we think it could be a very promising method to be introduced both in the cognitive and in the design phase of planning.

Round 2

Reviewer 3 Report

The manuscript had been revised accordingly. Please confirm the following two questions again.

1. There was only Table S2 but not Table S1 in the supplementary materials.

2. The sentence ( “The maps obtained from the overlay of the Graphic Cultivation Plans (GCPs) on LULC maps for the years 2016/2020 are separately converted into raster format, and then the average value of the weights taken by the pixels for the five years is calculated”) was on lines 456-459 instead of lines 661-670.